# Increased vaccine sensitivity of an emerging SARS-CoV-2 variant

Joseph A. Lewnard [1,2,3] ✉, Vennis Hong[4], Jeniffer S. Kim[4], Sally F. Shaw [4], Bruno Lewin[4,5], Harpreet Takhar[4], Marc Lipsitch[6] & Sara Y. Tartof [4,7] ✉

Host immune responses are a key source of selective pressure driving pathogen evolution. Emergence of many SARS-CoV-2 lineages has been associated with enhancements in their ability to evade population immunity resulting from both vaccination and infection. Here we show diverging trends of escape from vaccine-derived and infection-derived immunity for the emerging XBB/XBB.1.5 Omicron lineage. Among 31,739 patients tested in ambulatory settings in Southern California from December, 2022 to February, 2023, adjusted odds of prior receipt of 2, 3, 4, and ≥5 COVID-19 vaccine doses were 10% (95% confidence interval: 1–18%), 11% (3–19%), 13% (3–21%), and 25% (15–34%) lower, respectively, among cases infected with XBB/XBB.1.5 than among cases infected with other co-circulating lineages. Similarly, prior vaccination was associated with greater point estimates of protection against progression to hospitalization among cases with XBB/XBB.1.5 than among non-XBB/XBB.1.5 cases (70% [30–87%] and 48% [7–71%], respectively, for recipients of ≥4 doses). In contrast, cases infected with XBB/XBB.1.5 had 17% (11–24%) and 40% (19–65%) higher adjusted odds of having experienced 1 and ≥2 prior documented infections, respectively, including with pre-Omicron variants. As immunity acquired from SARS-CoV-2 infection becomes increasingly widespread, fitness costs associated with enhanced vaccine sensitivity in XBB/XBB.1.5 may be offset by increased ability to evade infection-derived host responses.

Host immune responses are a key source of selective pressure influencing the evolutionary dynamics of pathogens[1–3]. In the context of expanding population immunity, successive SARS-CoV-2 lineages have shown increasing capacity to evade both vaccine-derived and infection-derived immune responses throughout the course of the COVID-19 pandemic[4,5]. Whereas incremental reductions in protection against Epsilon, Gamma, Delta, and other early variants were generally found to be modest[6–9], the Omicron BA.1 lineage was associated with

≥1.8-fold higher risk of breaking through infection-derived immunity in comparison to Delta[10–13], as well as markedly lower effectiveness and duration of COVID-19 vaccine-derived protection[14–16]. Although epidemiologic studies did not find strong evidence of differential vaccine-derived or infection-derived protection against the BA.2 lineage in comparison to BA.1, the subsequent BA.4 and BA.5 lineages were associated with notably increased risk of re-infection relative to earlier Omicron lineages[17–20], as well as reduced vaccine effectiveness[21,22].

[1]Division of Epidemiology, School of Public Health, , University of California, Berkeley, Berkeley, CA 94720, USA. [2]Division of Infectious Diseases & Vaccinology, School of Public Health, University of California, Berkeley, Berkeley, CA 94720, USA. [3]Center for Computational Biology, College of Engineering, University of California, Berkeley, Berkeley, CA 94720, USA. [4]Department of Research & Evaluation, Kaiser Permanente Southern California, Pasadena, CA 91101, USA. [5]Department of Clinical Science, Kaiser Permanente Bernard J. Tyson School of Medicine, Pasadena, CA 91101, USA. [6]COVID-19 Response Team, Centers for Disease Control and Prevention, Atlanta, GA 30329, USA. [7]Department of Health Systems Science, Kaiser Permanente Bernard J. Tyson School of Medicine, Pasadena, CA 91101, USA. ✉e-mail: jLewnard@berkeley.edu; Sara.Y.Tartof@kp.org

Monitoring the ability of emerging lineages to evade immunity from vaccination or prior infection is central to ongoing efforts aimed at mitigating the burden of SARS-CoV-2, similar to experience with vaccines against influenza[23], pneumococcus[24], and other infectious disease agents[25,26].

The XBB/XBB.1.5 Omicron lineages emerged via recombination of BA.2.10.1 and BA.2.75 sublineages and overtook BQ.1/BQ.1.1, along with other BA.5-related lineages, as the leading cause of new infections within the US by late January 2023[27]. While XBB/XBB.1.5 evades neutralization by infection-derived antibodies[28,29], early observational studies have reported that updated (BA.4/BA.5-D614G bivalent) COVID-19 booster vaccination confers substantial protection against symptomatic XBB/XBB.1.5 infection[30]. It remains unclear whether XBB/XBB.1.5 differs from BA.5-related lineages in its sensitivity to host responses acquired through prior vaccination or infection. We therefore compared history of vaccination and documented SARS-CoV-2 infection, as well as clinical outcomes, among individuals infected with XBB/XBB.1.5 and co-circulating lineages derived from BA.4/BA.5 within the Kaiser Permanente Southern California (KPSC) healthcare system.

## Results
### Study population and setting
We analyzed data from 31,739 individuals who tested positive for SARS-CoV-2 infection in outpatient settings during the period from December 1, 2022, to February 23, 2023, within the KPSC healthcare system, an integrated care organization providing comprehensive medical services to ~4.7 million residents of southern California across outpatient, emergency department, inpatient, and virtual settings. We limited our sample to individuals whose specimens were processed using the ThermoFisher TaqPath COVID-19 combo kit (TF) assay (39.2% [31,739/80,894] of all eligible cases diagnosed during the study period; see Methods) in order to compare characteristics and clinical outcomes among cases whose positive tests included detection of the spike (S) gene probe or S-gene target failure (SGTF)—a well-described proxy for distinguishing XBB/XBB.1.5 and other BA.2-origin Omicron lineages (associated with S-gene detection) from other co-circulating lineages descending from BA.4 and BA.5 (associated with SGTF)[31]. Within a random sample of cases for whom sequencing results were available, the positive and negative predictive values of S-gene detection for ascertainment of XBB/XBB.1.5 infection and non-XBB/XBB.1.5 infection, respectively, were 98.2% (269/274) and 99.7% (1592/1597). Among sequenced specimens exhibiting SGTF, 99.2% (1585/1597) and 0.4% (7) were descendants of BA.5 and BA.4 lineages, respectively.

New detections of SARS-CoV-2 declined over the study period amid reductions in outpatient testing volume (Fig. 1). This decline in outpatient detections exceeded reductions in SARS-CoV-2 detection in inpatient settings, where all newly-admitted patients continued to be screened for SARS-CoV-2 infection at the point of admission throughout the study period. The proportion of TF-tested outpatient cases inferred to be infected with the XBB/XBB.1.5 lineage based on S-gene detection increased from 21.1% (45/213) as of December 1, 2022, to 77.8% (49/63) as of February 23, 2023. In total, our sample included 9869 cases infected with XBB/XBB.1.5 and 21,870 cases infected with other lineages. Characteristics of cases with XBB/XBB.1.5 and non-XBB/XBB.1.5 cases were similar, including their age, sex, and racial/ethnic distributions, as well as their prevalence of comorbid conditions and frequency of healthcare utilization over the preceding year (Table 1).

### Vaccination history among cases
Comparing vaccination and infection history among individuals infected with distinct lineages provides a strategy to identify differences in the protective effectiveness of these exposures against each lineage[32,33]. Importantly, this analytic approach may mitigate confounding that arises when comparing cases to uninfected controls[34], yielding estimates that indicate the relative degree of protection provided by vaccination or prior infection against each lineage. Within our sample, adjusted odds of having received 2, 3, 4, and ≥5 COVID-19 vaccine doses were 10% (95% confidence interval: 1–18%), 11% (3–19%), 13% (3–21%), and 25% (15–34%) lower among cases with XBB/XBB.1.5 in comparison to non-XBB/XBB.1.5 cases (Table 2), suggesting vaccine effectiveness against XBB/XBB.1.5 infection exceeded vaccine effectiveness against infection with co-circulating lineages. Analyses distinguishing vaccine doses received before or after individuals' first documented SARS-CoV-2 infection yielded similar findings (Supplementary Tables S1 and S2), as did analyses applying alternative frameworks to adjust for calendar time (Supplementary Table S3).

We next investigated whether these differences in vaccination status among cases with XBB/XBB.1.5 and non-XBB/XBB.1.5 cases could be explained by differences in timing of vaccination, or in receipt of bivalent doses, among cases infected with each lineage. However, these secondary analyses did not identify evidence that time since receipt of the last vaccine dose, or receipt of bivalent doses, differentially affected individuals' risk of infection with XBB/XBB.1.5 and other lineages. Adjusted odds of having received ≥4 vaccine doses >90 days prior to the date of testing were 17% (8–25%) lower among cases with XBB/XBB.1.5 than among non-XBB/XBB.1.5 cases (Supplementary

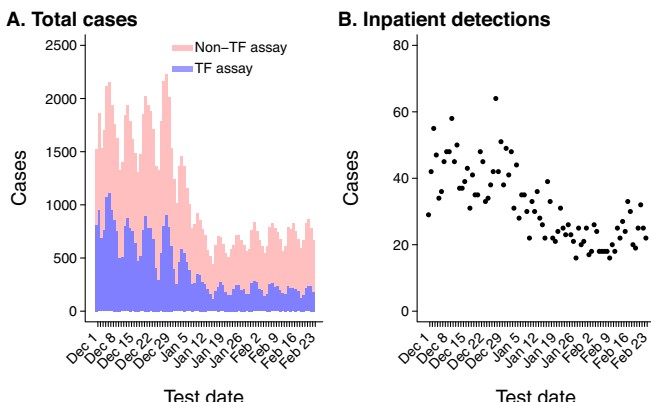

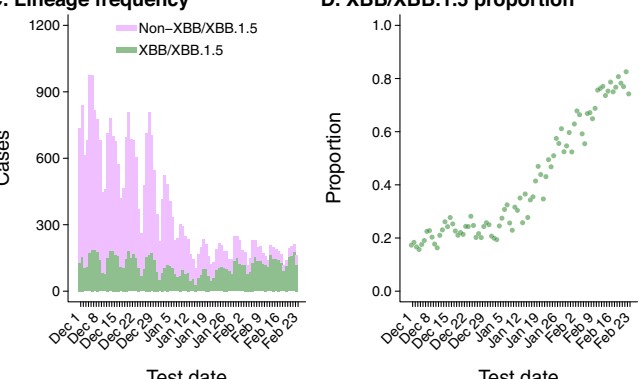

**Fig. 1 | Trends in SARS-CoV-2 infection diagnoses within the KPSC population.** Panels illustrate **A** frequency of SARS-CoV-2 detections via ThermoFisher TaqPath COVID-19 combo kit assay (TF; blue) and other assays (pink), across all settings; **B** frequency of SARS-CoV-2 detections among hospitalized patients; **C** frequency of samples with S-gene detection (green) or S-gene target failure (SGTF; violet), among outpatient cases tested via the TF assay; **D** proportion of outpatient cases with S-gene detection, among all positive outpatient cases tested with the TF assay. For consistency with panel (**C**), the proportion of isolates with S-gene detection (XBB/XBB.1.5 lineages) is illustrated in green in panel (**D**). Data include 80,894 cases, among whom 31,739 were tested using the TF assay and are represented in panels (**C**) and (**D**).

**Table 1 | Characteristics of outpatient COVID-19 cases with S-gene target failure or S-gene detection**

| Attribute | Count, n (%) | |
|---|---|---|
| | SGTF (non-XBB lineage) | S-gene detected (XBB lineage) |
| | N = 21,870 | N = 9869 |
| **Age (years)** | | |
| 0–9 | 923 (4.2) | 457 (4.6) |
| 10–19 | 1283 (5.9) | 638 (6.5) |
| 20–29 | 2307 (10.5) | 1043 (10.6) |
| 30–39 | 4015 (18.4) | 1661 (16.8) |
| 40–49 | 4066 (18.6) | 1676 (17.0) |
| 50–59 | 4105 (18.8) | 1888 (19.1) |
| 60–69 | 2962 (13.5) | 1402 (14.2) |
| 70–79 | 1647 (7.5) | 790 (8.0) |
| ≥80 | 562 (2.6) | 314 (3.2) |
| **Sex** | | |
| Female | 12,124 (55.4) | 5651 (57.3) |
| Male | 9746 (44.6) | 4218 (42.7) |
| **Charlson comorbidity index** | | |
| 0 | 14,643 (67.0) | 6469 (65.5) |
| 1–2 | 5411 (24.7) | 2511 (25.4) |
| 3–5 | 1313 (6.0) | 652 (6.6) |
| ≥6 | 503 (2.3) | 237 (2.4) |
| **Prior-year healthcare interactions** | | |
| 0–4 outpatient visits | 5472 (25.0) | 2375 (24.1) |
| 5–9 outpatient visits | 5340 (24.4) | 2551 (25.8) |
| 10–14 outpatient visits | 3679 (16.8) | 1621 (16.4) |
| 15–19 outpatient visits | 2312 (10.6) | 1121 (11.4) |
| 20–29 outpatient visits | 2520 (11.5) | 1091 (11.1) |
| ≥30 outpatient visits | 2547 (11.6) | 1110 (11.2) |
| Any ED presentation | 4773 (21.8) | 2220 (22.5) |
| Any inpatient admission | 1093 (5.0) | 465 (4.7) |
| **Race** | | |
| White, non-Hispanic | 4480 (20.5) | 1878 (19.0) |
| Black, non-Hispanic | 1745 (8.0) | 791 (8.0) |
| Hispanic | 11,278 (51.6) | 5351 (54.2) |
| Asian | 2828 (12.9) | 1176 (11.9) |
| Pacific Islander | 186 (0.9) | 79 (0.8) |
| Other/unknown/mixed race | 1353 (6.2) | 594 (6.0) |
| **Body mass index**[a] | | |
| Underweight (<18.5) | 879 (4.0) | 433 (4.4) |
| Normal weight (18.5–24.9) | 4036 (18.5) | 1812 (18.4) |
| Overweight (25.0–29.9) | 5574 (25.5) | 2538 (25.7) |
| Obese (30–39.9) | 6446 (29.5) | 2893 (29.3) |
| Severely obese (≥40.0) | 1592 (7.3) | 737 (7.5) |
| Missing | 3343 (15.3) | 1456 (14.8) |
| **Cigarette smoking**[a] | | |
| Current smoker | 831 (3.8) | 369 (3.7) |
| Former smoker | 3588 (16.4) | 1589 (16.1) |
| Never smoked | 14,421 (65.9) | 6712 (68.0) |
| Missing | 3030 (13.9) | 1199 (12.1) |
| **Neighborhood median household income**[a] | | |
| <$30,000 | 146 (0.7) | 53 (0.5) |
| $30,000–59,999 | 5145 (23.5) | 2438 (24.7) |
| $60,000–89,999 | 6946 (31.8) | 3068 (31.1) |
| $90,000–119,999 | 3912 (17.9) | 1688 (17.1) |

**Table 1 (continued) | Characteristics of outpatient COVID-19 cases with S-gene target failure or S-gene detection**

| Attribute | Count, n (%) | |
|---|---|---|
| | SGTF (non-XBB lineage) | S-gene detected (XBB lineage) |
| | N = 21,870 | N = 9869 |
| $120,000–149,999 | 885 (4.0) | 442 (4.5) |
| ≥$150,000 | 379 (1.7) | 161 (1.6) |
| Missing | 4457 (20.4) | 2019 (20.5) |

*SGTF* S-gene target failure, defined as cycle threshold readings of >37 for the S-gene and ≤37 for N and orf1a/b genes.
[a]Missing data imputed for regression analyses.

Table S4). Similarly, adjusted odds of having received ≥4 vaccine doses, but with ≥1 dose received <90 days before testing, were 15% (3–25%) lower among cases infected with XBB/XBB.1.5 than among non-XBB/XBB.1.5 cases. Consistent with these observations, distributions of times from receipt of the most recent vaccine dose were similar for cases with XBB/XBB.1.5 and non-XBB/XBB.1.5 cases. As compared to non-XBB/XBB.1.5 cases, cases with XBB/XBB.1.5 had 13% (1-23%) lower adjusted odds of having received a vaccine series with ≥4 monovalent doses only, and 18% (9-26%) lower adjusted odds of having received mixed monovalent/bivalent series with ≥4 doses (Supplementary Table S5). Neither the most recent vaccine product received nor the sequences of vaccine products received differed appreciably among XBB/XBB.1.5 cases and non-XBB/XBB.1.5 cases (Supplementary Tables S6 and S7).

Taken together, these observations suggest that although COVID-19 vaccination was associated with greater degrees of protection against XBB/XBB.1.5 infection than infection with other co-circulating lineages, the added benefits of recent boosting[35–37], including with bivalent vaccine doses[30], may not differ appreciably for protection against XBB/XBB.1.5 as compared to co-circulating lineages.

**History of documented SARS-CoV-2 infection among cases**
In a pattern opposite to these observations, cases with XBB/XBB.1.5 had 17% (11–24%) and 40% (19–65%) higher adjusted odds of having experienced 1 and ≥2 prior documented infections, respectively, in comparison to non-XBB/XBB.1.5 cases (Table 2). We undertook additional analyses aiming to determine whether these findings were explained by recent BA.4/BA.5 infection, which may have provided enhanced and specific protection against closely related lineages co-circulating with XBB/XBB.1.5 during the study period (Supplementary Table S8). Consistent with this hypothesis, cases with XBB/XBB.1.5 had 67% (40–99%) higher adjusted odds than non-XBB/XBB.1.5 cases of prior documented infection during the BA.4/BA.5 waves (June 25 to November 30, 2022), but did not have substantially higher adjusted odds of prior documented infection during the BA.1/BA.2 waves (December 20, 2021, to June 24, 2022; 4% [–3% to 12%] higher adjusted odds). However, cases with XBB/XBB.1.5 also had higher adjusted odds of prior documented infection during periods predominated by pre-Omicron variants, including the Delta variant (18% [3–35%]), Alpha/Epsilon variants (19% [4–36%]), and earlier lineages (29% [18–41%]). These findings suggested that protection associated with recent BA.4/BA.5 infection could only partially account for the observed association of XBB/XBB.1.5 infection with prior documented infection; higher odds of prior infection with pre-Omicron variants among cases with XBB/XBB.1.5 suggest that XBB/XBB.1.5 effectively evades infection-derived immunity associated with other SARS-CoV-2 variants as well.

We also undertook several sensitivity analyses aiming to verify that associations of XBB/XBB.1.5 with prior infection were not an artifact of suboptimal vaccine response among certain recipients.

**Table 2 | Association of S-gene detection with prior documented infection and COVID-19 vaccination**

| Exposure | Count, n (%) | | Odds ratio (95% CI) | | |
|---|---|---|---|---|---|
| | SGTF (non-XBB lineage) N = 21,870 | S-gene detected (XBB lineage) N = 9869 | Unadjusted | Time-adjusted[a] | Adjusted[b] |
| Prior infection | | | | | |
| 0 previous infections | 15,212 (69.6) | 6377 (64.6) | ref. | ref. | ref. |
| 1 previous infection | 6180 (28.3) | 3181 (32.2) | 1.23 (1.17, 1.29) | 1.17 (1.10, 1.24) | 1.17 (1.11, 1.24) |
| ≥2 previous infections | 478 (2.2) | 311 (3.2) | 1.55 (1.34, 1.80) | 1.39 (1.18, 1.63) | 1.40 (1.19, 1.65) |
| COVID-19 vaccination[c] | | | | | |
| 0 doses | 2704 (12.4) | 1314 (13.3) | ref. | ref. | ref. |
| 1 dose (any) | 499 (2.3) | 245 (2.5) | 1.01 (0.86, 1.19) | 1.01 (0.84, 1.21) | 1.01 (0.84, 1.22) |
| 2 doses (any) | 4648 (21.3) | 2065 (20.9) | 0.91 (0.84, 0.99) | 0.91 (0.83, 1.00) | 0.90 (0.82, 0.99) |
| 3 doses (any) | 8487 (38.8) | 3632 (36.8) | 0.88 (0.82, 0.95) | 0.91 (0.84, 0.99) | 0.89 (0.81, 0.97) |
| 4 doses (any) | 3664 (16.8) | 1720 (17.4) | 0.97 (0.89, 1.05) | 0.94 (0.85, 1.03) | 0.87 (0.79, 0.97) |
| ≥5 doses (any) | 1868 (8.5) | 893 (9.0) | 0.98 (0.89, 1.09) | 0.85 (0.76, 0.95) | 0.75 (0.66, 0.85) |

*SGTF* S-gene target failure, defined as cycle threshold readings of >37 for the S-gene and ≤37 for N and orf1a/b genes.

[a]Time-adjusted estimates are obtained via models defining intercepts for calendar week only.

[b]Adjusted estimates are obtained via models adjusted for calendar week, age (10-year bands), sex, race/ethnicity, current or former cigarette smoking, body mass index, Charlson comorbidity index, neighborhood socioeconomic status, and prior-year healthcare utilization across outpatient, inpatient, and emergency department settings. Covariates are categorized as listed in Table 1. Primary analyses exclude prior infection from the adjustment set for the association of vaccination with infecting lineage (Supplementary Fig. S1); estimates closely resemble results when prior infection is included in the adjustment set (Supplementary Table S12). Analyses separately considering vaccine doses received before or after individuals' first documented SARS-CoV-2 infection are presented in Supplementary Table S2. To investigate whether infections occurring after vaccination may serve as an indicator of poor vaccine response, prohibiting measurement of the direct effect, we also tested for associations of infecting lineage with immunocompromised or immunosuppressed status (Supplementary Table S10), but did not find strong evidence that cases with XBB/XBB.1.5 were more or less likely to have compromised or suppressed immune status in comparison to non-XBB/XBB.1.5 cases. Analyses defining calendar time continuously are presented in Supplementary Table S3.

[c]All vaccine types (BNT162b2, mRNA-1273, Ad.26.COV2.S, and NVX-CoV2373) are included. Frequencies of differing vaccine dose sequences are presented in Supplementary Table S7.

**Table 3 | Association of prior documented infection or vaccination with risk of hospital admission, among cases with S-gene detection or S-gene target failure**

| Exposure | SGTF (non-XBB lineage) | | | S-gene detected (XBB lineage) | | |
|---|---|---|---|---|---|---|
| | Count, n (rate per 100,000 person-days) | Time-adjusted hazard ratio (95% CI)[a] | Adjusted hazard ratio (95% CI)[b] | Count, n (rate per 100,000 person-days) | Time-adjusted hazard ratio (95% CI)[a] | Adjusted hazard ratio (95% CI)[b] |
| By infection history | | | | | | |
| No prior documented infection | 111 (24.8) | ref. | ref. | 49 (28.4) | ref. | ref. |
| 1 prior documented infection | 49 (28.4) | 1.02 (0.72, 1.44) | 0.93 (0.65, 1.33) | 46 (25.3) | 0.75 (0.44, 1.29) | 0.74 (0.41, 1.32) |
| ≥2 prior documented infections | 4 (28.4) | 1.12 (0.41, 3.03) | 0.72 (0.26, 2.01) | 2 (24.1) | 0.86 (0.21, 3.56) | 0.73 (0.17, 3.15) |
| By vaccination history | | | | | | |
| 0 doses | 20 (25.1) | ref. | ref. | 10 (28.0) | ref. | ref. |
| 1 dose[c] | 2 (13.6) | – | – | 1 (15.3) | – | – |
| 2 doses | 35 (25.6) | 1.01 (0.58, 1.75) | 0.94 (0.54, 1.65) | 11 (19.5) | 0.70 (0.30, 1.65) | 0.59 (0.24, 1.44) |
| 3 doses | 51 (20.4) | 0.81 (0.48, 1.35) | 0.54 (0.31, 0.93) | 23 (23.3) | 0.81 (0.39, 1.70) | 0.46 (0.21, 1.00) |
| ≥4 doses | 53 (32.7) | 1.27 (0.76, 2.13) | 0.52 (0.29, 0.93) | 24 (34.7) | 1.23 (0.59, 2.57) | 0.30 (0.13, 0.70) |

*SGTF* S-gene target failure, defined as cycle threshold readings of >37 for the S-gene and ≤37 for N and orf1a/b genes.

[a]Time-adjusted estimates are obtained via models defining strata for calendar week only.

[b]Adjusted estimates are obtained via models adjusted for calendar week, age (10-year bands), sex, race/ethnicity, current or former cigarette smoking, body mass index, Charlson comorbidity index, neighborhood socioeconomic status, and prior-year healthcare utilization across outpatient, inpatient, and emergency department settings. Covariates are categorized as listed in Table 1.

[c]Estimates for recipients of 1 dose are not presented due to sparse sample size. Analyses defining 0 or 1 dose receipt as the reference exposure yield the following adjusted hazard ratio estimates: among non-XBB/XBB.1.5 cases, 1.02 (0.60–1.77), 0.59 (0.35–0.99), and 0.57 (0.32–1.00) for 2, 3, and ≥4 doses, respectively, as compared to 0–1 doses; among XBB/XBB.1.5 cases, 0.70 (0.30–1.66), 0.54 (0.26–1.15), and 0.36 (0.16–0.81) for 2, 3, and ≥4 doses, respectively, as compared to 0–1 doses.

First, we distinguished prior infections acquired after individuals had received ≥2 COVID-vaccine doses from prior infections occurring among individuals who had received only a single dose or who were never vaccinated. Whereas post-vaccination infections could be an indicator of poor vaccine response, the same would not be true of pre-vaccination infections (Supplementary Fig. S1). Within these analyses, cases with XBB/XBB.1.5 had 23% (14–31%) higher adjusted odds of having experienced pre-vaccination infection as well as 13% (5–22%) higher adjusted odds of having experienced post-vaccination infections in comparison to non-XBB/XBB.1.5 cases (Supplementary Table S9). Further, we did not identify evidence of associations between infecting lineage and immunocompromised or immunosuppressed status (adjusted odds ratio 0.95 [0.89–1.05]), which would be expected to mediate differences in risk of post-vaccination infection among vaccine recipients (Supplementary Table S10). These findings suggest the enhanced capacity of the XBB/XBB.1.5 lineage to overcome

**Table 4 | Association of S-gene detection with risk of adverse clinical outcomes**

| Endpoint | Count, *n* (rate per 100,000 person-days) | | Hazard ratio (95% CI) | |
|---|---|---|---|---|
| | SGTF (non-XBB lineage) | S-gene detected (XBB lineage) | Time-adjusted[a] | Adjusted[b] |
| Hospital admission[c] | 161 (25.0) | 69 (25.9) | 1.02 (0.76, 1.38) | 1.03 (0.76, 1.38) |
| ICU admission[c] | 9 (0.7) | 7 (1.6) | 1.38 (0.47, 4.04) | 1.45 (0.51, 4.13) |
| Ventilation[c] | 1 (0.01) | 2 (0.05) | – | – |
| Death[c] | 0 | 0 | – | – |

*SGTF* S-gene target failure, defined as cycle threshold readings of >37 for the S-gene and ≤37 for N and orf1a/b genes.
[a]Time-adjusted estimates are obtained via models defining intercepts for calendar week only.
[b]Adjusted estimates are obtained via models adjusted for calendar week, age (10-year bands), sex, race/ethnicity, current or former cigarette smoking, body mass index, Charlson comorbidity index, neighborhood socioeconomic status, prior vaccination, prior documented infection, and prior-year healthcare utilization across outpatient, inpatient, and emergency department settings. Covariates are categorized as listed in Table 1.
[c]Endpoints include hospital admission due to any cause within 30 days after the index positive test; ICU admission due to any cause within 60 days after the index positive test; initiation of mechanical ventilation due to any cause within 60 days after the index positive test; and death due to any cause within 60 days after the index positive test.

immune responses associated with prior infection is independent of the observed association between prior vaccination and XBB/XBB.1.5 infection. In further confirmation of this finding, point estimates indicated stepwise increases in adjusted odds of having experienced 1 or ≥2 prior documented infections among cases with XBB/XBB.1.5 within strata encompassing recipients of 1, 2, 3, and ≥4 COVID-19 vaccine doses (Supplementary Table S11). These results support the directional relationship between infecting variant and history of documented SARS-CoV-2 infection identified in primary analyses.

### Predictors of disease progression

Protection against progression to severe disease endpoints provides an additional dimension for measuring how immunity from vaccination or prior infection impacts SARS-CoV-2 natural history[34,38]. Among cases with XBB/XBB.1.5 within our sample, prior receipt of 2, 3, and ≥4 COVID-19 vaccine doses was associated with 41% (−44% to 76%), 54% (0–79%), and 70% (30–87%) reductions, respectively, in adjusted hazards of progression to hospital admission due to any cause in the 30 days following a positive outpatient test (Table 3; single-dose effects were not estimated due to sparse counts). Corresponding estimates of protection against progression to hospital admission associated with 2, 3, and ≥4 COVID-19 vaccine doses were 6% (−65% to 46%), 46% (7–69%), and 48% (7–71%), respectively, among cases infected with other lineages. Our study was underpowered to estimate associations of vaccination with protection against progression to other severe disease endpoints including intensive care unit (ICU) admission, mechanical ventilation, or mortality; no deaths occurred among cases infected with either lineage over the course of follow-up (Table 4).

Documented prior infections were not strongly associated with risk of hospital admission among cases with XBB/XBB.1.5 or non-XBB/XBB.1.5 cases (adjusted hazard ratio for hospital admission after ≥2 documented infections equal to 0.73 [0.17–3.15] and 0.72 [0.26–2.01], respectively; Table 3). However, these associations are likely attenuated by misclassification of individuals' prior infection status. Infection with XBB/XBB.1.5 or non-XBB/XBB.1.5 lineage was not independently associated with individuals' likelihood of experiencing hospital admission or intensive care unit admission (adjusted hazard ratios equal to 1.03 [0.76–1.38] and 1.45 [0.51–4.13], respectively, for comparisons of cases with XBB/XBB.1.5 to non-XBB/XBB.1.5 cases; Table 4).

### Discussion

Cases in our study infected with the XBB/XBB.1.5 lineage had received fewer COVID-19 vaccine doses and had higher likelihood of prior documented SARS-CoV-2 infection, in comparison to contemporaneous cases infected with other circulating SARS-CoV-2 lineages. These findings suggest that although the XBB/XBB.1.5 lineage has greater capacity than co-circulating lineages (predominantly descending from BA.5) to evade immune responses triggered by prior infection, including with pre-Omicron variants, XBB/XBB.1.5 is more sensitive to immune responses triggered by COVID-19 vaccination. In further support of this hypothesis, COVID-19 vaccination was associated with greater point estimates of protection against hospital admission among cases with XBB/XBB.1.5 than among non-XBB/XBB.1.5 cases, although our sample size was insufficient to test for statistically significant differences in effect size estimates for this rare outcome. Reassuringly, our findings support the persistent benefit of vaccination for reducing individuals' risk of infection and severe disease with XBB/XBB.1.5, despite previously-reported immune-evasive characteristics of this lineage[28,29,39]. As capture of prior infections in individuals' electronic health records (EHRs) was likely incomplete for both cases with XBB/XBB.1.5 and non-XBB/XBB.1.5 cases, our study likely underestimates the true magnitude of association of prior infection with individuals' relative likelihoods of infection with XBB/XBB.1.5 or other SARS-CoV-2 lineages[40]. These results demonstrate the considerable ability of XBB/XBB.1.5 to evade immunity resulting from prior infection, in comparison to SARS-CoV-2 variants that emerged during earlier phases of the pandemic.

Our finding that enhanced vaccine sensitivity co-occurs with enhanced escape of infection-derived immunity in XBB/XBB.1.5 stands in contrast to observations of other SARS-CoV-2 variants emerging after widespread vaccine implementation. The BA.1 and BA.4/BA.5 Omicron lineages were associated with reductions in the protective effectiveness of both prior vaccination and infection in comparison to the Delta and BA.2 Omicron lineages they outcompeted[10-21]. Vaccine effectiveness against Delta variant infection was also mildly weaker than effectiveness against earlier variants[6,7], although emergence and dissemination of the Delta variant preceded widespread vaccination in most countries. While the greatest differences in lineage-specific vaccine protection have been apparent in effects on individuals' risk of acquiring infection, previous studies disaggregating vaccine effectiveness against infection from vaccine effectiveness against hospital admission[38] have demonstrated that vaccine escape may be associated with changes in protection against severe disease progression as well, consistent with findings in the present study. In early 2022, protection against progression from a positive outpatient test to hospital admission associated with prior receipt of ≥3 mRNA vaccine doses was 57% for cases within KPSC infected with the BA.1 Omicron lineage, versus 86% for cases infected with the Delta variant[10]. In the present study, receipt of 2, 3, and ≥4 COVID-19 vaccine doses was associated with greater stepwise increases in point estimates of protection against hospital admission for cases with XBB/XBB.1.5 (41%, 54%, and 70%, respectively) in comparison to non-XBB/XBB.1.5 cases (6%, 46%, and 48%, respectively). It is important to note that these estimates represent only the reduction in risk of severe disease progression among cases experiencing breakthrough infection following vaccination, and do not account for additional vaccine-derived protection against acquisition of SARS-CoV-2 infection—an important but separate

component of vaccine effectiveness against hospitalized illness[38]. Taken together, our observations are consistent with the hypothesis that XBB/XBB.1.5 shows enhanced sensitivity to vaccine-derived immune responses in comparison to other co-circulating lineages.

Immunological and evolutionary factors driving this apparent bifurcation in evasion of vaccine-derived and infection-derived responses for XBB/XBB.1.5 merit further investigation. Notably, vaccinations available in the US (mRNA-1273, BNT162b2, Ad.26.COV2.S, and NVX-CoV2373) target only the SARS-CoV-2 spike antigen. In contrast, infection with SARS-CoV-2 induces responses against an array of SARS-CoV-2 antigens, some of which may be independently associated with protection[41]. Among US blood donors, seroprevalence of infection-derived (anti-nucleocapsid) antibodies reached 58% by February 2022[42], prior to the widespread transmission of the BA.2 and BA.4/BA.5 variants. While US seroprevalence estimates following BA.4/BA.5 emergence are unavailable, studies in other countries have reported >80% prevalence of anti-nucleocapsid antibodies by late 2022, consistent with widespread transmission of these lineages[43,44]. As the prevalence of prior infection approaches or exceeds the proportion of individuals who have received all recommended primary series and booster doses of COVID-19 vaccines[27], the ability to evade responses against non-spike SARS-CoV-2 antigens might be of increasing importance to the fitness of emerging SARS-CoV-2 lineages. Enhanced infectivity, regardless of immune evasion, likewise contributes to the ability of novel SARS-CoV-2 lineages to achieve widespread transmission[8]. However, as prior studies have indicated that host receptor binding affinity of XBB/XBB.1.5 is weaker than that observed in co-circulating BQ.1/BQ.1.1 lineages[29,39], evasion of infection-derived immunity from both BA.4/BA.5 and pre-Omicron lineages may of greater relative importance to the successful establishment of XBB/XBB.1.5.

Our study has at least six limitations. First, our comparison of cases infected with XBB/XBB.1.5 and other co-circulating lineages is observational in nature. Although cases with XBB/XBB.1.5 and non-XBB/XBB.1.5 cases did not differ appreciably in most demographic characteristics or measured risk factors, unmeasured differences between the two case populations remain possible sources of confounding, and demographic or epidemiologic characteristics of the study population may not be generalizable to cases in other settings. Relatedly, our study was restricted to cases who received molecular testing in clinical settings, who may be distinct from individuals who received point-of-care antigen testing or who did not seek testing for their infections. However, selecting on receipt of a positive molecular test result enabled us to mitigate potential sources of confounding present in other study designs comparing infected cases to uninfected controls; fewer factors would be expected to confound comparisons of cases with XBB/XBB.1.5 and non-XBB/XBB.1.5 cases[34]. As our study conditions on individuals' acquisition of SARS-CoV-2 infection, observed associations should be interpreted as measures of the relative effect of vaccination and prior infection on individuals' risk of acquiring XBB/XBB.1.5, compared to their risk of acquiring other co-circulating lineages[45]. Alternative designs are needed to quantify the absolute effectiveness of prior infection and vaccination in preventing infection. Third, we relied on infections recorded in cases' EHRs to determine prior infection history. This is expected to result in exposure misclassification for both cases with XBB/XBB.1.5 and non-XBB/XBB.1.5 cases, as some prior infections may have gone undiagnosed or may have been diagnosed in settings outside the KPSC healthcare system. Our estimates should thus be interpreted as lower-bound measures of the association between prior infection and XBB/XBB.1.5 or non-XBB/XBB.1.5 infecting lineage. Fourth, our analyses of the association of infecting lineage with prior infection during periods predominated by the transmission of BA.4/BA.5, BA.1/BA.2, Delta, and

earlier lineages are subject to differing degrees of misclassification associated with changes over time in access to both clinical and at-home tests. While these analyses reveal that immune escape is not limited to evasion of responses associated with recent BA.4/BA.5 infection, comparisons of effect size estimates for prior infections with differing lineages should be made with caution. Relatedly, patients received diverse COVID-19 vaccine series (although Ad.26.COV2.S and NVX-CoV2373 were rarely administered within our study population; Supplementary Table S7). Although our analysis did not identify evidence that infection with XBB/XBB.1.5 or non-XBB/XBB.1.5 lineages was associated with either specific vaccine products received (which may indicate differential protection) or with time since vaccination (which may indicate differential waning), it is important to note these measures may not fully capture variation in vaccine-associated immune protection. Fifth, our study used all-cause hospital admissions to indicate severe disease progression. As some admissions may occur due to causes unrelated to COVID-19 in both cases with XBB/XBB.1.5 and non-XBB/XBB.1.5 cases, associations of vaccination with protection against disease progression may be underestimated. However, excluding cases tested in hospital settings helped to mitigate bias under our study design, as routine SARS-CoV-2 testing on admission may identify substantial numbers of incidental COVID-19 admissions related to screening[46,47] within studies including hospitalized patients. Last, our study includes individuals enrolled in KPSC health plans, among whom health status, socioeconomic status, and healthcare-seeking behavior may differ from the broader US population. While this indicates that measures from our study such as individuals' risk of hospital admission may not be externally generalizable, it is unlikely to affect the validity of observed associations between XBB/XBB.1.5 infection and prior infection or vaccination within the analytic sample.

Our analysis identifies increased vaccine sensitivity of the emerging SARS-CoV-2 XBB/XBB.1.5 lineage as well as an enhanced ability of this lineage to evade immunity associated with prior infection, including with pre-Omicron SARS-CoV-2 lineages. Selection for evasion of immune responses associated with prior infection, such as those targeting non-spike SARS-CoV-2 antigens, might be of growing importance to SARS-CoV-2 evolutionary trajectories as immunity from prior SARS-CoV-2 infection becomes increasingly prevalent; differences in COVID-19 vaccine coverage and prevalence of prior infection across settings may likewise become increasingly relevant to differences in the ability of XBB/XBB.1.5 and future lineages to establish circulation. While it is reassuring that prior vaccination is associated with enhanced protection against the XBB/XBB.1.5 lineage in comparison to co-circulating lineages, ongoing escape of infection-derived immunity remains a cause for concern. Continuous monitoring of changes in protection associated with vaccination and prior infection is needed to inform responses to emerging SARS-CoV-2 variants.

## Methods
### Study setting
We undertook this retrospective observational study within the KPSC healthcare system. As a comprehensive, integrated care organization, KPSC delivers healthcare across telehealth, outpatient, emergency department, and inpatient settings for ~4.7 million members enrolled through employer-provided, government-sponsored, and pre-paid coverage schemes. EHRs across all clinical settings, together with laboratory, pharmacy, and immunization data, provide a complete view of care delivered by KPSC. These observations are augmented by insurance claims for out-of-network diagnoses, prescriptions, and procedures, enabling near-complete capture of healthcare interactions for KPSC members. The KPSC Institutional Review Board reviewed and provided ethical approval for the study.

During the study period, roughly 15% of all outpatient cases with SARS-CoV-2 infection confirmed by molecular testing received nirmatrelvir-ritonavir, although this population accounted for only ~25% of all nirmatrelvir-ritonavir prescribing (as the majority of patients received treatment on the basis of clinical or at-home antigen test results). Although also available during the study period, molnupiravir was rarely used (<0.1% of all cases), as it was reserved for patients unable to receive nirmatrelvir-ritonavir due to potential drug-drug interactions[48].

## Eligibility criteria

We included cases who: (1) received a positive molecular test result in any outpatient setting between December 1, 2022, and February 23, 2023; (2) had specimens processed using the ThermoFisher TaqPath COVID-19 Combo Kit (as described below); (3) had not received any prior SARS-CoV-2 test result in a clinical setting or prior COVID-19 diagnosis within 90 days before their index test; (4) had been enrolled in KPSC health plans for ≥1 year prior to their index test, allowing for up to a 45-day lapse in membership to account for potential delays in re-enrollment; and (5) were not hospitalized at the time of their index test, and had not been hospitalized at any point in the preceding 7 days. Excluding individuals without ≥1 year of membership in KSPC health plans (N = 3749 out of 35,488 otherwise eligible cases) enabled us to ensure COVID-19 vaccine doses, infection history, comorbid conditions, and healthcare utilization were captured accurately for the analytic sample. Restricting analyses to outpatient cases was expected to provide several design advantages. First, this strategy helped to ensure cases infected with each lineage were similar to each other in terms of healthcare-seeking behavior[34,49]. Second, initiating follow-up from the point of outpatient testing helped to ensure cases were ascertained at similar stages of their clinical course, facilitating unbiased comparisons of subsequent progression to severe disease. Outpatient-diagnosed cases at KPSC were automatically enrolled in a home-based symptom monitoring program with standardized criteria for emergency department referral and inpatient admission as a measure to preserve hospital capacity throughout the study period[50]. Thus, hospital admission was considered to represent an internally consistent measure of disease severity within the sample followed from the point of outpatient testing[10], whereas cases first intercepted in hospital settings may have had greater variation in clinical status at the point of testing; in the event that infections were acquired in the hospital, these cases may also have differed with respect to clinical status, SARS-CoV-2 exposure, and time to treatment initiation in comparison to those acquiring infection in the community. This approach also helped our study to avoid the inclusion of incidental SARS-CoV-2 detections among patients who were tested at the point of inpatient admission for causes unrelated to COVID-19. Last, whereas the TaqPath COVID-19 Combo Kit was the primary assay used at regional reference laboratories for outpatient testing, cases diagnosed in hospital settings may have had tests processed in-house using other assays, without readout enabling lineage determination.

## Lineage calling

The TaqPath COVID-19 Combo Kit assay included probes for the spike (S), nucleocapsid (N), and orf1a/b genes. Cases with cycle threshold ($c_T$) values below 37 for ≥2 probes were considered positive for SARS-CoV-2. We interpreted S-gene target failure (SGTF), defined as $c_T \geq 37$ for the S-gene but $c_T < 37$ for the N and orf1a/b genes, as a proxy for infection with BA.4/BA.5 sublineages, whereas S-gene detection provided a proxy for infection with XBB/XBB.1.5 lineages, consistent with our validation data (Results) and prior US studies[30,31].

## Exposures

We characterized the following attributes of included cases using data from their EHRs: age (defined in 10-year age bands), biological sex; race/ethnicity (white, black, Hispanic of any race, Asian, Pacific Islander, and other/mixed/unknown race, as self-reported by individuals and recorded in their medical record); neighborhood socioeconomic status, measured as the median household income within their Census block (<$30,000, $30,000–59,999, $60,000–89,999, $90,000–119,999, $120,000–149,999, and ≥$150,000 per year); cigarette smoking status (current, former, or never smoker); body mass index (BMI; categorized as underweight, normal weight, overweight, obese, or severely obese); Charlson comorbidity index (0, 1–2, 3–5, and ≥6); prior-year emergency department visits and inpatient admissions (each defined as 0, 1, 2, or ≥3 events); prior-year outpatient visits (0–4, 5–9, 10–14, 15–19, 20–29, or ≥30 events); documented prior SARS-CoV-2 infection; and history of COVID-19 vaccination (receipt of 0, 1, 2, 3, 4, or ≥5 doses, according to manufacturer, type, and time from receipt of each dose to the date of the index test). While our analyses include recipients of BNT162b2, mRNA-1273, Ad.26.COV2.S, and NVX-COV2373, we do not distinguish protection associated with receipt of mRNA or non-mRNA vaccine series, as non-mRNA vaccine doses accounted for only 2.0% of all vaccine doses received within the study population (1741/86,076). For individuals with multiple prior COVID-19 diagnoses or positive SARS-CoV-2 test results, we considered these infections to be distinct if they were not preceded by any other COVID-19 diagnosis or positive test result within 90 days. For data anonymization, index test dates were jittered by random addition of −1, 0, or 1.

## Outcomes

We followed outpatient-diagnosed cases from the point of their index test through death, disenrollment, or March 5, 2023, the date of the database cut (providing ≥10 days of follow-up for individuals who did not die or disenroll). The primary endpoint was hospital admission for any cause within 30 days after the index test. Additional endpoints monitored over 60 days after the index test included ICU admission, initiation of mechanical ventilation, and death.

## Multiple imputation of missing data

To accommodate missing data on cases' BMI (n = 4799; 15.1%), cigarette smoking (n = 4229; 13.3%), and neighborhood household income (n = 6476; 20.4%), we generated 10 pseudo-datasets completed by sampling from the conditional distribution of these variables, given all other observed characteristics of cases, via multiple imputation. We conducted complete-case statistical analyses across each of the 10 pseudo-datasets and pooled results across these analyses according to Rubin's rules[51].

## Logistic regression analysis

Within this analytic sample of cases testing positive for SARS-CoV-2 infection, potential outcomes were binary (infection with XBB/XBB.1.5 or non-XBB/XBB.1.5 lineages). We estimated adjusted odds ratios of prior vaccination and prior documented infection among cases with XBB/XBB.1.5 and non-XBB/XBB.1.5 cases via logistic regression. Models used in primary analyses controlled for variables as categorized in Table 1, as motivated by a directed acyclic graph (Supplementary Fig. S1), and included fixed intercepts for the calendar week of testing. As a sensitivity analysis, we also fit models defining calendar time continuously via polynomial transformations of individuals' calendar date of testing. Models including a fourth-order polynomial function were found to provide optimal penalized fit to data on the basis of minimizing the Bayesian Information Criterion (Supplementary Table S3).

For all analyses, we report unadjusted associations of each exposure with XBB/XBB.1.5 or non-XBB/XBB.1.5 infection, associations accounting only for testing week ("time-adjusted" odds ratios),

and associations accounting for all confounders ("adjusted" odds ratios). Models using alternative adjustment strategies for vaccination and infection (Supplementary Fig. S1) provided results similar to those of the primary analysis (Supplementary Tables S2, S8, S9, S11 and S12).

## Survival analysis

For analyses addressing the association of prior vaccination, infection, and infecting lineage with cases' risk of hospital admission or ICU admission, we fit Cox proportional hazards models to data on cases' times to each of these events or censoring (at study end date, end of follow-up at 30 or 60 days, or disenrollment, whichever occurred earliest). A survival analysis framework was motivated by the fact that XBB/XBB.1.5 infections accounted for an increasing share of all diagnosed cases over time, and thus had higher likelihood of censoring within <30 days or <60 days in comparison to non-XBB/XBB.1.5 cases. Models defined strata according to cases' calendar week of testing to control for potential changes in testing and healthcare-seeking practices. We verified the proportional hazards assumption by testing for non-zero slopes of the Schoenfeld residuals[52].

## Software

We conducted analyses using R (version 4.0.3; R Foundation for Statistical Computing, Vienna, Austria). We used the survival[53] package (version 3.5-3) for time-to-event analyses, and the Amelia II package[54] (version 1.81.1) for multiple imputation.

## Reporting summary

Further information on research design is available in the Nature Portfolio Reporting Summary linked to this article.

# Data availability

Individual-level testing and clinical outcomes data reported in this study are not publicly shared due to privacy protections for patient electronic health records. Individuals wishing to access disaggregated data, including data reported in this study, should submit requests for access to the corresponding author (sara.y.tartof@kp.org). Requests will receive a response within 14 days. De-identified data (including, as applicable, participant data and relevant data dictionaries) will be shared upon approval of analysis proposals with signed data-access agreements in place.

# Code availability

Analysis code is available from GitHub (https://github.com/joelewnard/xbb)[55].

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

## Acknowledgements

This work was funded by the US Centers for Disease Control and Prevention (CDC; grant 75D3-121C11520 to S.Y.T.). The findings and conclusions in this report are those of the authors and do not necessarily represent the official position of the CDC. J.A.L. was supported by grant R01-AI14812701A1 from the National Institute for Allergy and Infectious Diseases (US National Institutes of Health), which had no role in the design or conduct of the study, or the decision to submit for publication.

## Author contributions

J.A.L., M.L., and S.Y.T. contributed to the study concept and design. V.H., J.S.K., S.F.S., B.L., H.T., and S.Y.T. led the acquisition of data. V.H. and J.A.L. led the statistical analysis of data. J.A.L., M.L., and S.Y.T. led the interpretation of data. J.A.L. drafted the manuscript, and all authors critically revised the manuscript for important intellectual content. S.Y.T. obtained funding and provided supervision.

## Competing interests

J.A.L. has received research grants paid directly to his institution and consulting honoraria unrelated to this study from Pfizer. S.Y.T. has received research grants paid directly to her institution unrelated to this study from Pfizer. The remaining authors disclose no competing interests.
