## [Peer Review File · Nature Communications]

Increased vaccine sensitivity of an emerging SARS-CoV-2 variantREVIEWER COMMENTS

Reviewer #1 (Remarks to the Author):

This is a well conducted, robust and helpful analysis of a large patient cohort in Southern California. The manuscript is well written, succinct and easy to follow, although I did get a little confused in a few places and I have the following questions/suggestions for the authors:

1. I think it would be useful to include a table (or other representation) of the number of individuals who have received each combination of vaccine given throughout the study. For example, how many individuals are there who have received 4 doses of BNT162b2 or 4 doses of mRNA-1273?
2. On a similar note, I think it is important to provide the vaccination status of the participants versus previous COVID infection status (for example, I could not work out how many of those with 0, 1, or 2+ COVID infections had received 0, 1, 2, 3 etc doses of COVID vaccine.
3. The authors should have the power to undertake a subgroup analysis by vaccine brand of last dose administered. This is a major relevant public health question and I think within the scope of this manuscript.
4. Is there any immune waning that can be observed from these data?

5. The footnote of Table 2 states 'The distribution of vaccine types received is as follows: 'BNT162b2 monovalent: 42,641 (9.5% of all doses received); BNT162b2 bivalent: 3,658 (4.2% of all doses received); mRNA-1273 monovalent: 35,557 (41.3% of all doses received); mRNA-1273 bivalent: 2,479 (2.9% of all doses received); Ad.26.COV2.S: 1,729 (2.0% of all doses received); NVXCoV2373: 12 (<0.1% of all doses received).'

I may have gotten confused here, but when I summed the percentages I got 60%, which doesn't make sense as the total of 'all doses.'

Please can the authors clarify what they mean in this footnote and amend it to either correct it or make it easier to understand for the reader.

6. Please can the authors explain why participants had to be enrolled in the KPSC health plan for >1 year before index case?
- If this is because other data (such as medical attendance) would be unavailable otherwise, could this not be handled in a different way and thus avoid excluding participants? Or perhaps a sensitivity analysis?

7. In the methods section, the authors state 'cases first intercepted in hospital settings may have had greater variation in clinical status at the point of testing'.

- Is there not also the potential for a different pressure of infection in a hospital setting vs OP setting, and different speed at treatment initiation etc, which might affect outcomes? - If so, this should also be mentioned in this manuscript.

8. Can the authors comment on whether the insurance/fee-paying in the US healthcare system is likely to have biased results in this analysis? How would this compare across and/or affect other healthcare system models?

Dr Catherine Hyams, Bsc(Hons) MBBS PhD, University of Bristol

Reviewer #2 (Remarks to the Author):

This is a well presented, careful and thorough analysis of observational healthcare data providing population-level epidemiological evidence of the relative protection afforded by vaccination and prior infection against XBB lineage as compared to non-XBB lineage SARS-CoV-2 infection. The study finds greater ability of the XBB lineage to evade infection-derived immunity but enhanced sensitivity to vaccine induced immunity.

I have only a couple of minor comments.

(-) Clearly there are strong effects of time at play. The proportion of XBB/XBB.1.5 infections is strongly associated with time over the ~3-month study period, as is # vaccine doses and previous infections necessarily. Might there be residual confounding left after adjustment for calendar week? It would be helpful to get a sense of how robust the OR estimates are to alternative handling of the temporal effects. More comprehensive adjustment might for example be achieved by fitting the index date through a parametric curve (you can do a lot with the number of df currently being spent on fitting separate indicator variables) or through conditional logistic regression.

(-) in table 1, it would be helpful to include an additional row for each of the BMI, smoking and income variables indicating the number with missing values in each lineage group

(-) more of a reflection than a reviewer comment: further to the study's finding of XBB's enhanced sensitivity to vaccine-induced immunity, I wonder if the relative success of XBB in the US compared to many European countries might in part be a result of lower vaccination coverage in the US?

Reviewer #3 (Remarks to the Author):

Lewnard et al. provided evidence of increased vaccine sensitivity as well as increased ability to evade infection-derived host responses specifically for XBB/XBB.1.5, using an appropriate dataset from the outpatient setting with molecular testing for S-gene detection. The analytic plan was well described and appropriate for the study aim. The authors have also carefully analyzed the interplay between prior vaccinations and infections. The findings are likely robust, further supported by relevant sensitivity and subgroup analyses.

Major comments

1. Could the authors comment on the use of oral antivirals over the study period that should have reduced hospital admissions?
2. Discussion, "we identify that COVID-19 vaccination is associated with greater degrees of protection against hospital admission among cases with XBB/XBB.1.5 than among non-XBB/XBB.1.5 cases". Table 3 shows that the adjusted HR was lower for XBB/XBB.1.5 but were that statistically significant differences?

Minor comments

3. Figure S1B. Could the authors comment on whether the causal relation is complete? Prior infection affects the timing of vaccination and could also affect the number of doses (I0 -> V).
4. Figure S1B. Could the authors clarify if the adjustment of I0 is needed for estimating the direct effect of I1 on Y?
5. "Infection with XBB/XBB.1.5 or non-XBB/XBB.1.5 lineage was not independently associated with individuals' likelihood of experiencing hospital admission" (Table 3) Vaccination should be well recorded but there were likely significant misclassifications using documented infection as a proxy to actual infections. The estimated protective effect against hospital admission from prior infections may have been attenuated. This has been noted in the limitations but could be made more clearly in the results or discussion.

REVIEWER COMMENTS

Reviewer #1 (Remarks to the Author):

This is a well conducted, robust and helpful analysis of a large patient cohort in Southern California. The manuscript is well written, succinct and easy to follow, although I did get a little confused in a few places and I have the following questions/suggestions for the authors:

We thank the Reviewer for this assessment and for the suggestions below.

1. I think it would be useful to include a table (or other representation) of the number of individuals who have received each combination of vaccine given throughout the study. For example, how many individuals are there who have received 4 doses of BNT162b2 or 4 doses of mRNA-1273?

We have added Table S7 disaggregating individuals' vaccination histories according to the specific sequence of booster doses (if any) received after a primary series of BNT162b2, mRNA-1273, or a single dose of Ad.26.COVS.2 or NVX-CoV2373.

2. On a similar note, I think it is important to provide the vaccination status of the participants versus previous COVID infection status (for example, I could not work out how many of those with 0, 1, or 2+ COVID infections had received 0, 1, 2, 3 etc doses of COVID vaccine.

We have added Table S1 disaggregating the number of infections individuals experienced and the number of vaccine doses received.

3. The authors should have the power to undertake a subgroup analysis by vaccine brand of last dose administered. This is a major relevant public health question and I think within the scope of this manuscript.

We have added the requested analysis in Table S6; analyses address differences according to the last dose administered. While we do not identify any noteworthy differences with respect to vaccine type, we value this suggested addition to the study.

4. Is there any immune waning that can be observed from these data?

While our case-only design does not address the "absolute" effectiveness of any vaccine sequence or timing, the framework allows us to determine whether the duration of protection conferred by vaccination wanes differentially for one lineage in comparison to the other. We have revised the Discussion to better clarify this interpretation of the findings related to time since vaccination; our findings suggest waning is not differential by lineage (lines 245-249).

5. The footnote of Table 2 states 'The distribution of vaccine types received is as follows: BNT162b2 monovalent: 42,641 (9.5% of all doses received); BNT162b2 bivalent: 3,658 (4.2% of all doses received); mRNA-1273 monovalent: 35,557 (41.3% of all doses received); mRNA-1273 bivalent: 2,479 (2.9% of all doses received); Ad.26.COVS.2: 1,729 (2.0% of all doses received); NVXCoV2373: 12 (<0.1% of all doses received).'

I may have gotten confused here, but when I summed the percentages I got 60%, which doesn't make sense as the total of 'all doses.'

Please can the authors clarify what they mean in this footnote and amend it to either correct it or make it easier to understand for the reader.

We have removed the footnote and instead refer readers to the newly-added Table S7, which we believe presents vaccine histories (in terms of not only vaccine types but sequences of vaccine types) in a clearer manner.

6. Please can the authors explain why participants had to be enrolled in the KPSC health plan for >1 year before index case?

- If this is because other data (such as medical attendance) would be unavailable otherwise, could this not be handled in a different way and thus avoid excluding participants? Or perhaps a sensitivity analysis?

We required enrollment in a KPSC health plan for ≥ 1 year to ensure that relevant comorbid conditions, histories of healthcare utilization, and vaccination and infection status were captured in individuals' healthcare record; we are unaware of alternative frameworks for capturing such information if individuals were not enrolled in KPSC health plans. However, this exclusion criterion had only a small bearing on the

final population captured in analyses; dropping this would result in an analytic sample of 35,488 individuals instead of 31,739 individuals. We consider this “cost” acceptable given the value of having comprehensive information on exposures including comorbidities, healthcare utilization, and history of vaccination and infection among all individuals analyzed. We have clarified these points in the revised Methods (lines 282-286).

7. In the methods section, the authors state ‘cases first intercepted in hospital settings may have had greater variation in clinical status at the point of testing’.

- Is there not also the potential for a different pressure of infection in a hospital setting vs OP setting, and different speed at treatment initiation etc, which might affect outcomes? - If so, this should also be mentioned in this manuscript.

We have amended the manuscript to include these additional considerations (lines 305-307).

8. Can the authors comment on whether the insurance/fee-paying in the US healthcare system is likely to have biased results in this analysis? How would this compare across and/or affect other healthcare system models?

We have added further points on external generalizability to the Discussion (lines 254-258).

Dr Catherine Hyams, Bsc(Hons) MBBS PhD, University of Bristol

Reviewer #2 (Remarks to the Author):

This is a well presented, careful and thorough analysis of observational healthcare data providing population-level epidemiological evidence of the relative protection afforded by vaccination and prior infection against XBB lineage as compared to non-XBB lineage SARS-CoV-2 infection. The study finds greater ability of the XBB lineage to evade infection-derived immunity but enhanced sensitivity to vaccine induced immunity.

We thank the Reviewer for this assessment and for the suggestions below.

I have only a couple of minor comments.

(-) Clearly there are strong effects of time at play. The proportion of XBB/XBB.1.5 infections is strongly associated with time over the ~3-month study period, as is # vaccine doses and previous infections necessarily. Might there be residual confounding left after adjustment for calendar week? It would be helpful to get a sense of how robust the OR estimates are to alternative handling of the temporal effects. More comprehensive adjustment might for example be achieved by fitting the index date through a parametric curve (you can do a lot with the number of df currently being spent on fitting separate indicator variables) or through conditional logistic regression.

We agree the parametric curve approach may offer value and avert the need to fit so many intercepts. We have added this analysis in Table S3; based on the Bayesian Information Criterion, we selected models defining 4th-order functions of cases' calendar dates of testing (see Methods, lines 351-354 and Table S3 caption). However, OR estimates and accompanying CI ranges are virtually unchanged from those of the primary analyses.

(-) in table 1, it would be helpful to include an additional row for each of the BMI, smoking and income variables indicating the number with missing values in each lineage group

We have added the number with missing data as a row for each of these variables.

(-) more of a reflection than a reviewer comment: further to the study's finding of XBB's enhanced sensitivity to vaccine-induced immunity, I wonder if the relative success of XBB in the US compared to many European countries might in part be a result of lower vaccination coverage in the US?

This is indeed an intriguing possibility—if true it provides a provocative ecological proof-of-concept skirting the potential sources of confounding present in analyses of individual-level data such as our study. While our study does not test this hypothesis specifically, we have added the point to the conclusions paragraph that differences in population immunity across settings (from prior infection and vaccination) may become increasingly relevant to the ability of variants to become established (lines 264-266).

Reviewer #3 (Remarks to the Author):

Lewnard et al. provided evidence of increased vaccine sensitivity as well as increased ability to evade infection-derived host responses specifically for XBB/XBB.1.5, using an appropriate dataset from the outpatient setting with molecular testing for S-gene detection. The analytic plan was well described and appropriate for the study aim. The authors have also carefully analyzed the interplay between prior vaccinations and infections. The findings are likely robust, further supported by relevant sensitivity and subgroup analyses.

We thank the Reviewer for this assessment and for the suggestions below, which we believe have helped us to improve the manuscript.

Major comments

1. Could the authors comment on the use of oral antivirals over the study period that should have reduced hospital admissions?

We have clarified (lines 282-286) that by the time of the study, ~15% of outpatient cases with SARS-CoV-2 infection confirmed by molecular testing were receiving nirmatrelvir-ritonavir, although this accounted for a small proportion of all prescribing as the majority of patients received this treatment on the basis of rapid antigen tests (such patients are not represented in this study). Molnupiravir, although also available, was rarely used (<0.1% of cases) as providers were instructed to prescribe nirmatrelvir-ritonavir preferentially, and (when possible) to temporarily withhold or modify dosage of other medications for which drug-drug interactions could be of concern so that patients could receive nirmatrelvir-ritonavir.

2. Discussion, “we identify that COVID-19 vaccination is associated with greater degrees of protection against hospital admission among cases with XBB/XBB.1.5 than among non-XBB/XBB.1.5 cases”. Table 3 shows that the adjusted HR was lower for XBB/XBB.1.5 but were that statistically significant differences?

We have added the caveat that the sample size was not sufficient to test formally for a significant difference in effect size estimates (lines 173-175). For greater clarity on this point, we have revised the main text to state that “point estimates” differed in all parts of the manuscript referring to this finding.

Minor comments

3. Figure S1B. Could the authors comment on whether the causal relation is complete? Prior infection affects the timing of vaccination and could also affect the number of doses ($I_0 \rightarrow V$).

We recognize it is a limitation that vaccination status is a time-varying construct (as is infection history) and that representing “V” as a single variable in the DAG is a simplification. We have revised the caption text to clarify that other representations of V are possible, including distinguishing vaccination courses received after a known infection (which could be impacted by I_0) from those received before a known infection (which could not be impacted by I_0); in this event, it would be necessary to condition on post-infection vaccinations to estimate the effect of I_0 on Y. This does correspond to an analysis we have undertaken; Table S2 disaggregates individuals’ vaccination status as doses received before or after infection, and we have pointed this out in the revised caption.

While the state space of immunological statuses individuals may occupy (accounting for all potential sequences of infection and vaccination) is large, we believe the analyses presented in the Main Text and supplement (accounting for numbers of infections and vaccine doses as well as timing of infection before vs. after vaccination, and vaccination before vs. after infection) provide a broad exploration of the key considerations and areas where effect modification may occur.

4. Figure S1B. Could the authors clarify if the adjustment of I_0 is needed for estimating the direct effect of I_1 on Y?

If one defines an independent effect of I_0 on Y (bottom curved arrow in Figure S1B) then adjustment for I_0 is necessary to estimate the direct effect of I_1 on Y; this is the basis for the analysis presented in Table S9 where we separately estimate effects of I_0 and I_1 . If one instead hypothesizes that there is no meaningful immunological difference between infections occurring before or after vaccination, then the DAG collapses to that presented in Figure S1A and the primary analysis Results (Table 2) apply.

5. “Infection with XBB/XBB.1.5 or non-XBB/XBB.1.5 lineage was not independently associated with individuals’ likelihood of experiencing hospital admission” (Table 3) Vaccination should be well recorded but there were likely significant misclassifications using documented infection as a proxy to actual infections. The estimated protective effect against

hospital admission from prior infections may have been attenuated. This has been noted in the limitations but could be made more clearly in the results or discussion.

We appreciate the suggestion to better highlight this issue. So that it is at the forefront of readers' attention when they read about this analysis, we have raised this point in the Results where these data are first presented (lines 160-161).

REVIEWERS' COMMENTS

Reviewer #1 (Remarks to the Author):

Thank you for making the request changes/points of clarifications and providing the additional analyses. I think that the paper is much clearer, and agree that even though there is no differential effect by vaccine brand this is a useful analysis.

Reviewer #2 (Remarks to the Author):

The authors have done a good job of addressing all reviewer suggestions. I have no further comments